# Synergistic growth of nickel and platinum nanoparticles via exsolution and surface reaction

Min Xu [1,6], Yukwon Jeon[2,6], Aaron Naden [1], Heesu Kim[2], Gwilherm Kerherve[3], David J. Payne [3,4], Yong-gun Shul [5] & John T. S. Irvine [1]

Bimetallic catalysts combining precious and earth-abundant metals in well designed nanoparticle architectures can enable cost efficient and stable heterogeneous catalysis. Here, we present an interaction-driven in-situ approach to engineer finely dispersed Ni decorated Pt nanoparticles (1-6 nm) on perovskite nanofibres via reduction at high temperatures (600-800 °C). Deposition of Pt (0.5 wt%) enhances the reducibility of the perovskite support and promotes the nucleation of Ni cations via metal-support interaction, thereafter the Ni species react with Pt forming alloy nanoparticles, with the combined processes yielding smaller nanoparticles that either of the contributing processes. Tuneable uniform Pt-Ni nanoparticles are produced on the perovskite surface, yielding reactivity and stability surpassing 1 wt.% Pt/$\gamma$-Al$_2$O$_3$ catalysts for CO oxidation. This approach heralds the possibility of in-situ fabrication of supported bimetallic nanoparticles with engineered compositional distributions and performance.

Precious metals, especially Pt, find wide industrial applications, typically as efficient nanoparticle catalysts dispersed on supports[1-4], however, it is essential to reduce the usage of precious metals and improve their activity and stability[5,6]. Incorporating earth-abundant transition metals to precious metals is known to achieve this goal in rational design strategies for applications such as heterogeneous catalysis[7-10]. Furthermore, the catalytic properties of monometallic materials can be modified by a second metal through effects such as interfacial or lattice strain effects[11-14]. Generally, the fabrication of bimetallic supported catalysts uses combined deposition onto a backbone support[15-17]. It is quite challenging to control the morphology, homogeneity and support interaction of bimetallic nanoparticles at a support surface with the limited flexibility of simple a deposition procedure.

Recently, an in-situ approach to the growth of metal nanoparticles has been developed, referred to as exsolution from a host material, such as perovskite materials, driven by reduction conditions, as shown in Fig. 1a[18]. Research groups have synthesised monometallic noble metal particles, such as Pt[19], Rh[20], Pd[21], Ru[22], and Ir[23], decorated on perovskite hosts through exsolution for environmental and renewable energy applications. However, the nanoparticles were relatively large in size (>5 nm) and hence the catalytic properties were not optimal[24,25]. Thus, designing efficient catalysts with the desired composition, size and structure remains a significant challenge.

Here, we report an interaction-driven Pt-Ni nanoparticle synthesis on perovskite nanofibres through an in-situ emergent procedure as shown in Fig. 1a. The emergence of the Ni species from the doped perovskite was facilitated by the monodispersed Pt species coating on the fibre surface, which stabilised the growth of the nanoparticles to below 5 nm with a tenth of the size of Ni particles exsolved at the same condition. In turn, the emergence of Ni from the backbone affords a dynamic formation of bimetallic nanoparticles, rendering great catalytic properties by unique metal-support interactions.

[1]School of Chemistry, University of St Andrews, St Andrews, UK. [2]Department of Environmental and Energy Engineering, Yonsei University, Wonju, Republic of Korea. [3]Department of Materials, Imperial College London, London, UK. [4]Research Complex at Harwell, Harwell Science and Innovation Campus, Didcot, Oxfordshire, UK. [5]Department of Chemical and Biomolecular Engineering, Yonsei University, Wonju, Republic of Korea. [6]These authors contributed equally: Min Xu, Yukwon Jeon. ✉ e-mail: jtsi@st-andrews.ac.uk

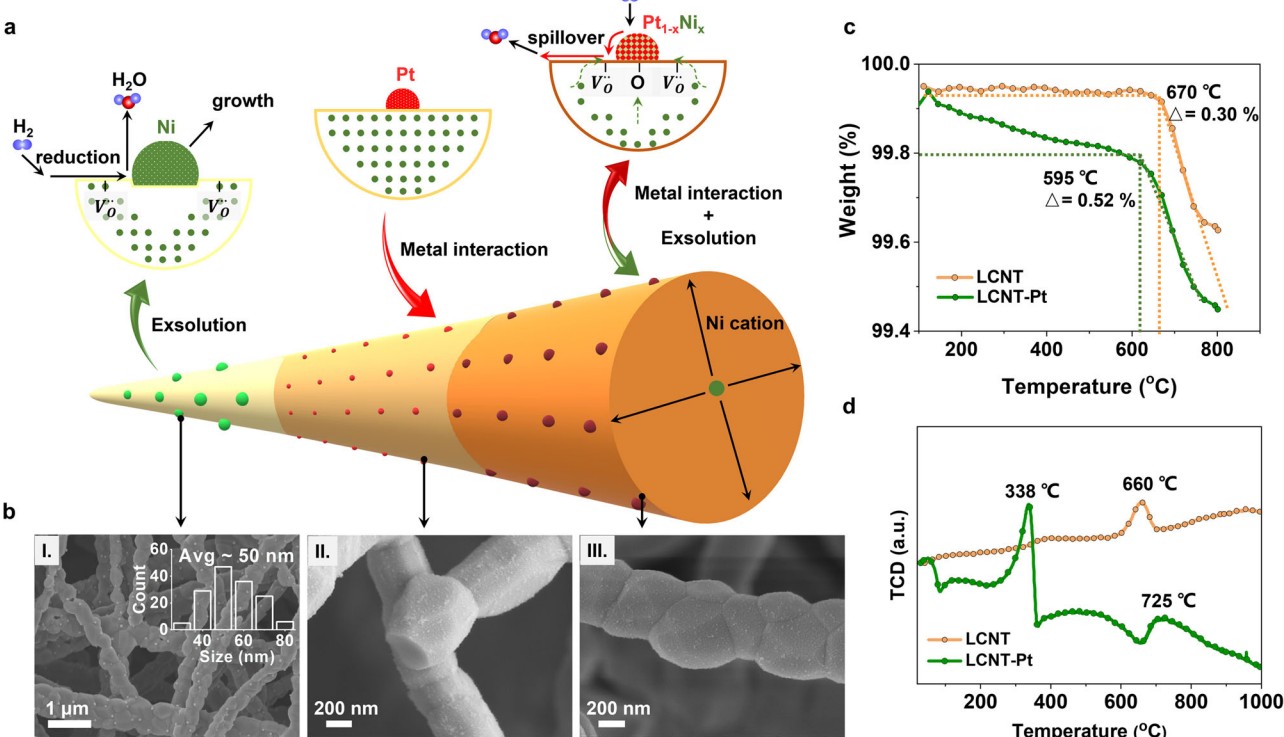

**Fig. 1 | Catalyst design with in-situ Ni decoration under Pt nanoparticles on perovskite. a** Schematic images comparing catalyst fabrication processes. **b** SEM micrograph of corresponding catalysts with $La_{0.52}Ca_{0.28}Ni_{0.06}Ti_{0.9}O_3$ fibres. I. LCNT fibres reduced at 800 °C for 4 h in 5% $H_2$/Ar (inset: size distribution of Ni particles), II. Pt deposited on LCNT fibres and treated at 400 °C in air to decompose nitrates, III. LCNT-Pt fibres reduced 600 °C for 4 h in 5% $H_2$/Ar. **c** Thermogravimetric analysis plot of weight as a function of temperature step and (**d**) Temperature programmed reduction profiles of LCNT and Pt deposited LCNT fibres.

## Results

### Catalyst design and fabrication

Studies focused on a non-stoichiometric Ni-doped perovskite $La_{0.52}Ca_{0.28}Ni_{0.06}Ti_{0.94}O_3$ (LCNT) with 0.2 deficiency on A-site, which has been recently reported to promote B-site dopant emergence[24]. This perovskite with a nanofibrous architecture was investigated as a platform for supported catalysts, prepared by an electrospinning method (Supplementary Fig. 1). The perovskite nanofibres were produced with a high aspect ratio (diameter to length) with a length of more than 100 μm and an average diameter of about 350 nm. This morphology was maintained over the differently treated samples as confirmed in Fig. 1b (also see in Supplementary Fig. 1b, c). Nanoparticles with an average size of about 50 nm were exsolved on the unmodified nanofibres after reducing in 5% $H_2$/Ar for 4 h at 800 °C as shown by scanning electron microscopy (SEM) (Fig. 1b I, denoted as LCNT-800R).

Pt modification was then considered; Pt nanoparticles can be particularly effective when applied as very small-sized particles in supported catalysts[26]. Pt nitrate precursor infiltrated nanofibres were calcined at 400 °C for 2 h in the air to decompose the nitrate, depositing well-dispersed Pt nanoparticles on the pristine nanofibre surface with a very low loading of 0.5 wt% (See support information and Supplementary Table 1). The uniform web structure allowed efficient precursor diffusion resulting in a fine distribution of small nanoparticles (Fig. 1b II). No obvious X-ray powder diffraction (XRD) lines for Pt species were detected (Supplementary Fig. 2), in accord with the low loading. The prepared samples were subsequently reduced at higher temperature to trigger the formation of Ni exsolution under the deposited Pt, resulting in smaller nanoparticles decorating the perovskite fibres surface. Thermogravimetric analysis (TGA) shows more extensive and facile weight loss under hydrogen atmosphere for the sample with Pt loading than the unloaded perovskite (Fig. 1c).

The onset weight loss point shifts to lower temperature by about 70 °C, with about 40 % more weight loss for the Pt loaded samples until the temperature reached 800 °C. This shows more oxygen vacancy formation (0.111 vs 0.085 on reduction at 800 °C for 10 h) under the same treatment conditions most likely due to enhanced surface exchange due to the presence of Pt (see Supplementary Fig. 3 and related discussion).

The fast hydrogen spillover from Pt to perovskite may help pump out the lattice oxygen in oxide support[27]. The adsorption properties were probed for perovskite oxide with and without Pt by temperature programmed reduction (TPR). The transient hydrogen uptake during the heating phase of the TPR shows a pronounced hydrogen uptake peak around 665 °C for the Pt-free sample contributing to the bulk reduction (Fig. 1d). However, the presence of Pt on perovskite yielded an apparent change in TPR around 338 °C, suggesting the reduction of Pt and surface oxygen from the support oxide[28]. The deposited Pt nanoparticles facilitate fast dissociation of hydrogen, therefore, accelerating the stripping of lattice oxygen from the support oxide. This is consistent with previous reports where the deposited metal increased the oxide host reducibility[29].

The SEM micrograph in Fig. 1b shows well-dispersed and much smaller nanoparticles decorated on the surface of perovskite fibres with Pt than without Pt (also see Supplementary Fig. 4 for comparison with other samples). With the same deposition procedure applied, small particles formed on the surface even for a pellet sample (Supplementary Fig. 5). The slightly larger particle size may relate to uneven wetting and differences in surface composition across the crystal orientations. This highlights the advantage to achieving more small nanoparticles on the fibre framework, where the naturally weaved network with high heat resistance ensures to confine the diffusion for decorated nanoparticles therefore prohibit the possible agglomeration even at high activation temperatures.

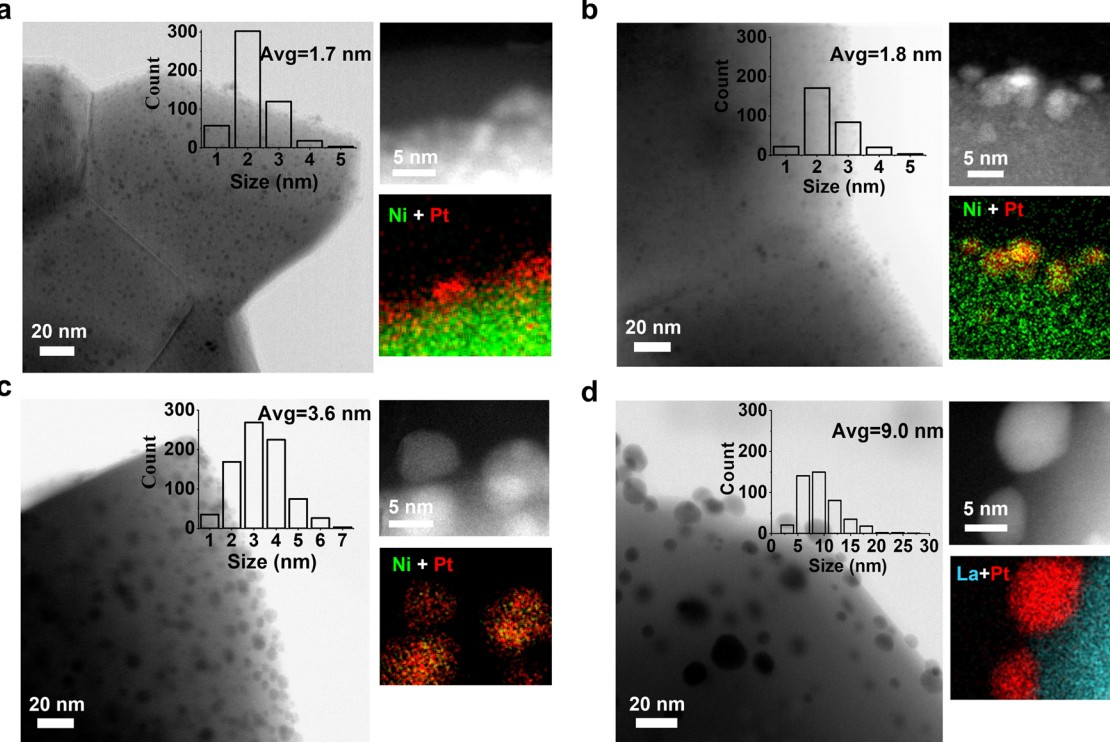

**Fig. 2 | Pt-(Ni) nanoparticles by Ni emergent from the tailored perovskite materials.** Representative STEM images and EDX elemental maps of the nano-particles for LCNT-Pt nanofibres reduced at (**a**) 400 °C, (**b**) 600 °C, (**c**) 800 °C for 4 h in 5 % H$_2$/Ar, and (**d**) LCT-Pt nanofibre reduced at 800 °C for 4 h in 5 % H$_2$/Ar. (inset: size distribution of nanoparticles, count on the corresponding STEM image).

To carefully evaluate the particles evolution process, the prepared samples were subsequently reduced at 400 °C, 600 °C and 800 °C (denoted as LCNT-Pt400R, LCNT-Pt600R and LCNT-Pt800R) for 4 h in 5% H$_2$/Ar. Figure 2 displays representative scanned transmission electron microscopy (STEM) images to track morphology changes of the nanoparticles size and distribution on the perovskite surface, and the corresponding energy dispersion X-ray spectroscopy (EDX) elemental maps for the nanoparticles compositions. For LCNT-Pt400R, the deposited nanoparticles were found in a size distribution from 1 to 4 nm with an average of 1.7 nm and highly dispersed over the surface, while LCNT-Pt600R possesses a similar size distribution with a slightly increased average diameter of 1.8 nm (Fig. 2a, b). In the case of LCNT-Pt800R bigger nanoparticles are observed with a mean diameter of around 3.6 nm (Fig. 2c), probably incorporating exsolved Ni at high temperature.

Even after heat treatment at high temperatures, the nanoparticles at LCNT-Pt800R are still well-distributed and are in the range of an appropriate size (<5 nm) for a highly effective catalytic activity[30]. Different to the observation of high-temperature reduction on titania-supported platinum where encapsulation of Pt NPs occurred, the surface energy minimisation was achieved by alloying with exsolution of Ni. Compared to the system without the Ni dopant (La$_{0.52}$Ca$_{0.28}$TiO$_3$, denoted as LCT), the nanoparticles displayed a wide size distribution with an average of 9 nm (Fig. 2d). The resistivity against nanoparticle coalescence of the Ni exsolution on the Pt nanoparticles size and distribution was also shown in Supplementary Fig. 6, by comparison with the Pt deposition onto an already reduced sample. As the exsolved Ni tends to be etched by the acidic Pt precursor solution, there are many pits created with unevenly dispersed Pt species remaining.

### In-situ Ni decoration Pt nanoparticles

To understand the underlying structure and formation mechanism of the nanoparticles, Fig. 3 shows high-resolution STEM (HR-STEM) analysis for compositional and structural characterisations. Cross-sections of the nanoparticle specimens were prepared from the LCNT-Pt nanofibres by focused ion beam (FIB) milling technique (Supplementary Figs. 7–10). As for the compositional analysis of LCNT-Pt600R in Fig. 3a, Supplementary Figs. 8 and 9, the EDX mapping image shows ~2 nm Pt alloy nanoparticles with a small but homogeneous Ni content distributed at the surface, with most Ni still residing in the bulk of perovskite. The nanoparticles are Pt enriched with the Pt:Ni ratio of about 4:1 (Supplementary Fig. 8, some of them with Pt:Ni ratio about 10:1 as shown in Supplementary Fig. 10). The formed Pt$_{1-x}$Ni$_x$ nanoparticles display a structure with Pt shell overlaying Ni as shown in the Fig. 3a and Supplementary Fig. 10b similar with previous reported Pt-skin catalyst which achieves a superior electrocatalytic performance[31]. When increasing the reduction temperature to 800 °C as shown in Fig. 3b, Supplementary Figs. 11 and 12, 3-4 nm nanoparticles on the perovskite surface with Pt-Ni alloyed features were verified by STEM-EDX mapping analysis. The elemental distribution across the nanoparticle shows a homogeneous compositional 1:1 ratio for the Pt:Ni content which is consistent with the EDX profile. The formation of mono-dispersed bimetallic nanoparticles on the surface, showing more Ni egressed from the host perovskite and incorporated into the particles.

Interestingly, the deposited Pt species on the perovskite clearly seem to facilitate the exsolution of Ni from the backbone. Compositional line-scan profiles across the nanoparticle and interface obtained through HAADF-STEM-EDX show a depleted Ni layer (about 1.4 nm as shown in the rectangular frame in Fig. 3a) formed in the underlying perovskite. The Ni depletion indicates the regions underneath the particles supply materials for the growth of nanoparticles before reaching an equilibrium state where a uniform compositional distribution is usually observed[32]. This suggests that the mounted Pt particles provide a nucleation site for the reduction cations from the support, consistent with observation on weight loss during reduction which indicates more defects formed (Fig. 1c). The doped Ni in the

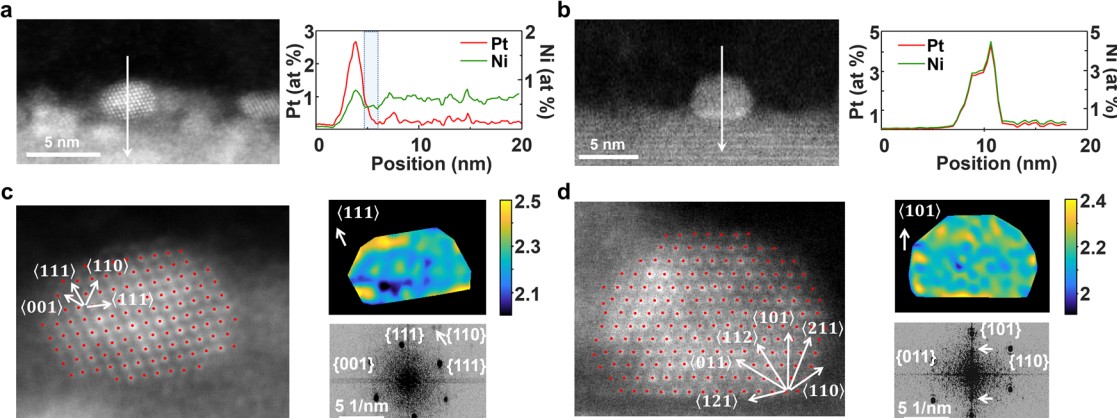

**Fig. 3 | Compositional and structural characterisations of the Ni decorated Pt nanoparticles.** HR-STEM analysis by line EDX of the nanoparticles corresponding to LCNT-Pt samples reduced at (**a**) 600 °C and (**b**) 800 °C, as well as the HR-STEM HAADF overlaid by Gaussian fitting, FFT images with lattice directions and strain distributions of the selected nanoparticles corresponding to LCNT-Pt samples reduced at (**c**) 600 °C (along <111> direction) and (**d**) 800 °C (along <101> direction, the reflections <100> of perovskite support were marked with yellow arrows), respectively.

lattice gradually exsolved to the surface region of the perovskite by extrinsic forces, which is generally offered by the high-temperature reduction atmosphere[33]. As the formation of Pt-Ni intermetallic compounds has the lowest formation enthalpy compared to other Pt-based intermetallic compounds like Pt-Ti[34], the Pt species on the perovskite surface facilitates the diffusion of $Ni^{2+}$ by the local metal-support coordination. Furthermore, our method provides an efficient formation of distinctive topmost layers terminated with Pt-Ni nanoparticles, while typically a high-temperature annealing, ultra-high vacuum or long growth time are applied to produce Pt-M alloy with nanosegregated Pt skins[9,35].

Investigation of the interfacial HR-STEM images for LCNT-Pt600R in Fig. 3c and Supplementary Fig. 13a displays poorly aligned, ~7° away from the perovskite <110> zone axis, and randomly oriented nanoparticles with the support, which indicates a weak interaction. The nanoparticles seated on LCNT-Pt800R (110) terminations are well aligned, just ~2° away from the perovskite zone axis, grown in epitaxial alignment to [110] native surface facet, as seen in Fig. 3d and Supplementary Fig. 13b. This keeps the growth of nanoparticles with a pseudo-cubic structure in a continuous way by an anchoring effect from Ni exsolution[33]. The HR-STEM images also depict the truncated Wulff shape of Pt-Ni nanoparticles on the (100) fringe of perovskite, which is the equilibrium shape of a supported nanocrystal related to the surface free energy and interfacial energy[36].

Close investigation of the atomic resolutions of the nanoparticles at the HAADF STEM images in Fig. 3c, d was carried out by measuring the atomic column positions (overlaid red spot with pixel accuracy) using Gaussian fitting. The measured lattice fringes of nanoparticles for LCNT-Pt600R are in the plane of (001), (111) and (110), which was also assigned from the FFT image. The d-spacing at the (111) plane was calculated to be 2.27 ± 0.09 Å from the deviation in FFT image (Supplementary Table 2), close to the d-spacing of Pt(111) enriched bimetallic nanoparticles[37–39]. After 800 °C reduction, the d-spacing in Supplementary Table 3 shows a single-crystal structure (P4/mmm) with well-defined fringes of Ni-Pt alloy, in excellent agreement between measured and bulk values. From an additional calculation, a structural mismatch of 56 % was found in direction [101] at the nanoparticle and LCNT (3.87 Å) interface, which generates lattice strains in the Ni-Pt alloy nanoparticles.

Atomic resolution images also provide a deviation map from the variation in d-spacing of the particle for a strain distribution in the nanoparticles[40], as shown in Fig. 3c, d and Supplementary Fig. 13. As the strain in a nanoparticle can be intrinsic due to the finite size[41], the Pt nanoparticle on LCNT-Pt600R present strains in the particles due to

the partial combination with Ni, and displays non-uniform distributions. The blurred FFT reflections also support this conclusion, where uneven distribution indicates a local/disorder strain of the nanoparticles. The nanoparticles on LCNT-Pt800R are different with evenly distributed additional strains mainly located on the corners/edges due to a lattice mismatch at the nanoparticle–support interface. These localised strains generally modify the chemical properties of the metallic systems[40], which can afford an improved catalytic property. From the detail microscopy examinations, through control of releasing transition metals by structural design in perovskite systems, a dynamic tuning of the alloy nanoparticle atomic structure can be possible to regulate the catalytic performance.

**Chemical structure during dynamic restructuring**

Figure 4 displays X-ray absorption spectroscopy (XAS) and X-ray photoelectron spectroscopy (XPS) measurements that provide the evolution of oxidation states and local coordination of each chemical bonding structure. As indicated in Fig. 4a, the X-ray absorption near-edge structure (XANES) spectra for Ni K-edge displayed a steep edge and high white line at 8350 eV, consistent with $Ni^{2+}$. The height of the white line decreased by increasing the reduction temperature from 600 °C towards the spectral feature of the Ni foil (metallic $Ni^0$), while Ni content from XPS analysis increased at the perovskite surface (Supplementary Fig. 14). The XPS results of the Ni 2p and LMM reveal increased intensity of $Ni^{2+}$ $2p_{1/2}$ and metallic Ni for the LCNT-Pt800R due to the accelerated Ni diffusion to the surface[42]. The Ni local structure was directly observed, corresponding to Fourier transforms of the extended X-ray-absorption fine-structure (EXAFS) oscillations for Ni K-edge (Fig. 4b and Supplementary Fig. 15). As expected, Ni-O coordination at 1.62 Å mainly exists for all as-prepared samples. On the other hand, peaks at the position of around 2.2 Å, which represents the first Ni-Ni coordination shell contribution, were raised in LCNT-Pt600R and a highly mixed Ni oxide and metallic phase at LCNT-Pt800R was exhibited, which corresponds to the TGA results (Fig. 1c). The Ni-Ni peak for LCNT-Pt800R is slightly shifted to the left, indicating an alloying effect from the interaction with the loaded Pt on the perovskite surface. This is also consistent with the fitting results where the coordination number of the Ni-Ni increased from 2.4 to 5.8 with the raised reduction temperature with an opposite tendency for the Ni-O (Supplementary Table 4).

The Pt $L_3$-edge XANES spectra display a similar trend with Ni for the reduced samples (Fig. 4c and Supplementary Fig. 16). As the Pt is found to be mainly in a metallic phase of $Pt^0$, the white-line intensities nearby 11,580 eV, links to a transition from $2p_{3/2}$ to 5d, decreased in the

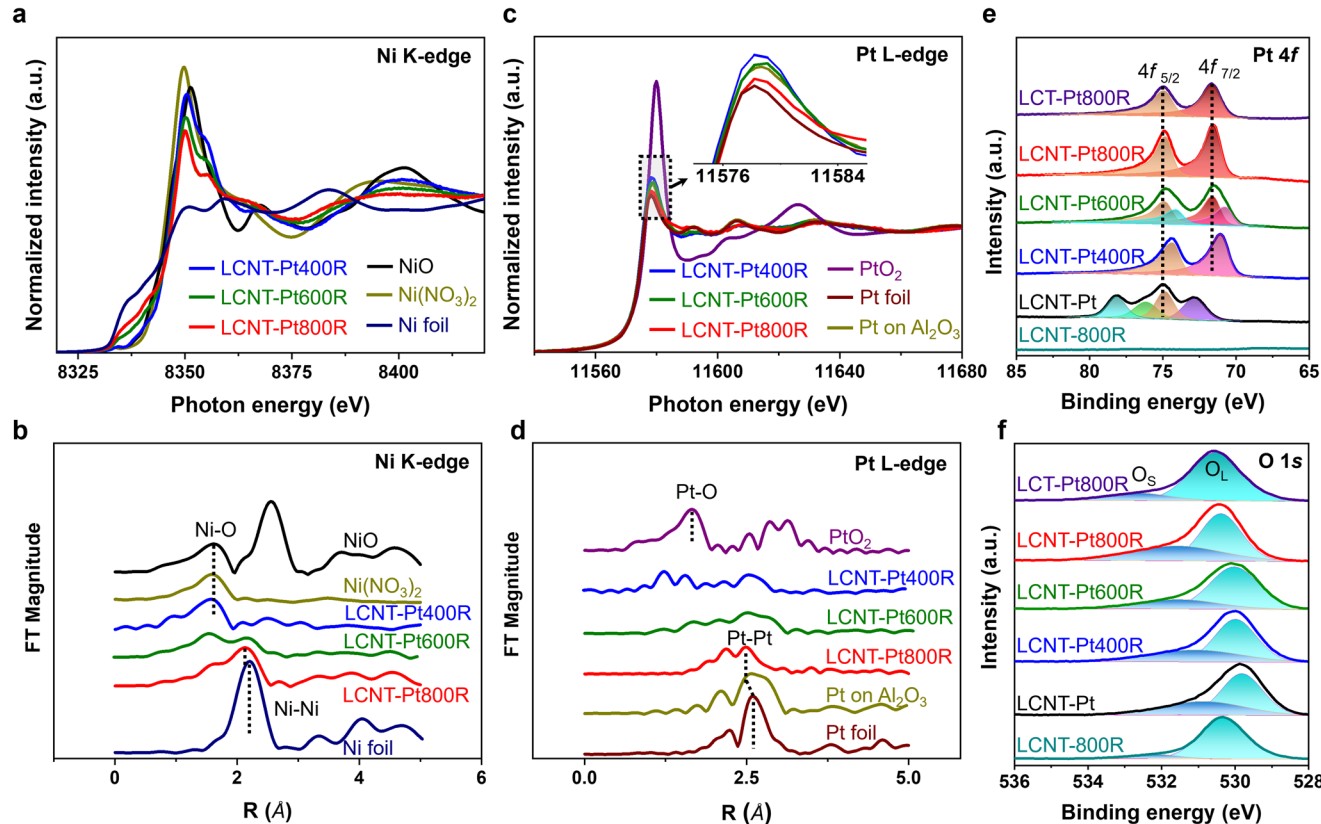

**Fig. 4 | Chemical characterisations of the Ni decorated Pt nanoparticles on perovskite fibres. a** Ni K-edge XANES spectra, (**b**) Ni K edge EXAFS Fourier–transform spectra, (**c**) Pt L$_3$-edge XANES spectra, (**d**) Pt L$_3$-edge EXAFS Fourier–transform spectra (Ni foil, NiO, Ni(NO$_3$)$_2$ for Ni K edge and Pt foil, PtO$_2$, Pt/Al$_2$O$_3$ for Pt L$_3$-edge as reference samples), and XPS analysis of (**e**) Pt 4 f and (**f**) O 1 s regions (LCNT and LCT-Pt fibres reduced at 800 °C as background references), corresponding to the samples of LCNT-Pt fibres subsequently reduced at 400 °C, 600 °C, and 800 °C. O$_L$ and O$_S$ are lattice oxygen and surface oxygen species, respectively.

order of 800 °C < 600 °C < 400 °C reductions, which specifies the decrease in the unoccupied $d$ states due to the charge transfer of Pt-Ni alloy and/or Pt to the support[43]. Therefore, a further decrease in Pt $d$-band vacancies, than the Pt on γ-Al$_2$O$_3$, is anticipated with a high electron density that can also mean a change to a positively charged Pt species, resulting in a reduction of the intermediates adsorptive strength. The radial structures for the first Pt-O and Pt-Pt coordination shells function around Pt atoms were calculated by EXAFS analysis. Figure 4d indicates mainly metallic Pt species from the relatively strong contribution of Pt-Pt bonding (~ 2.7 Å) for the reduced samples, while LCNT-Pt400R shows a mixed structure with Pt-O bonding (~ 1.8 Å) due to less reduction. Close examination of Pt-Pt shells for the treated samples in Fig. 4d reveals a reduction in intensities, which can explain a structural disordering of Pt atoms by metal-support interaction. The consistent changing of coordination number of the Pt-Pt and Pt-O during the reduction suggests an interaction between the Pt and the support (Supplementary Fig. 17 and Supplementary Table 4). The alteration of the Debye-Waller factor (σ$^2$, the mean square deviation of interatomic distances) is opposite for Ni and Pt, indicates the different structural ordering during the interaction. Furthermore, the slight shift of the edge positions is identical to that of the Pt foil by a formation of the strong bimetallic state for LCNT-Pt800R. This feature was also detected by XPS analysis in Fig. 4e, which displays a negative shift in the binding energy of two distinct (fitted) peaks of Pt 4$f_{7/2}$ (71.5 eV) and 4$f_{5/2}$ (74.9 eV) core levels[44,45], attributed to the alloying of Pt with Ni consistent with the microscopy and XAFS results.

XPS O 1 $s$ shows a gradual peak shift as reduced temperature increased for Pt deposited samples (Fig. 4f). These peaks were fitted according to two distinguishable species for lattice oxygen and chemisorbed oxygen[46,47], where the relative percentages were calculated in Supplementary Table 5. The higher ratio of surface oxygen species for Pt decorated samples indicates more lattice oxygen was driven away from the support oxide, which is consistent with the TGA and TPR results (Fig. 1c and Fig. 1d). XPS Ti 2$p$ shift to higher energy after reduced at high temperature while the nickel doped samples exhibit less energy disparity compared to LCT-Pt, as shown in Supplementary Fig. 14d. On this basis, it is appreciable that apical Pt assists the B-site cations out, mainly facilitates the Ni exsolution by extracting lattice oxygen through catalytic activation from interaction to adjust the perovskite structure with more surface oxygen for efficient oxidation systems.

## Catalytic performance studies

CO oxidation studies have been performed, Fig. 5, to probe the degree of catalytic CO oxidation activity and stability of these nanoparticles. The observed activation energy for catalytic reactions at supported nanoparticles is highly dependent on the nature of the active metal and perovskite host, as well as the interaction between them[30]. As expected, Fig. 5a shows that the perovskite fibre without Pt and reduced at 800 °C, demonstrated low catalytic activity with a light off at 370 °C due to a catalytic limitation of the Ni metal and its particle size of over 50 nm. By the synergistic growth of nanoparticles with Pt and Ni, the co-loaded samples show good catalytic performance from shifts in light-off temperatures and for example TOF$_{150}$ values of 0.053 and 0.189 s$^{-1}$ for LCNT-Pt-400R and LCNT-Pt-600R, respectively (Supplementary Fig. 18a). Amongst the Pt containing samples, LCNT-Pt800R fibre exhibited a superior performance with a $T_{99}$ at 130 °C at low Pt loading (0.5 wt.%) for CO conversion, Fig. 5a. This catalytic

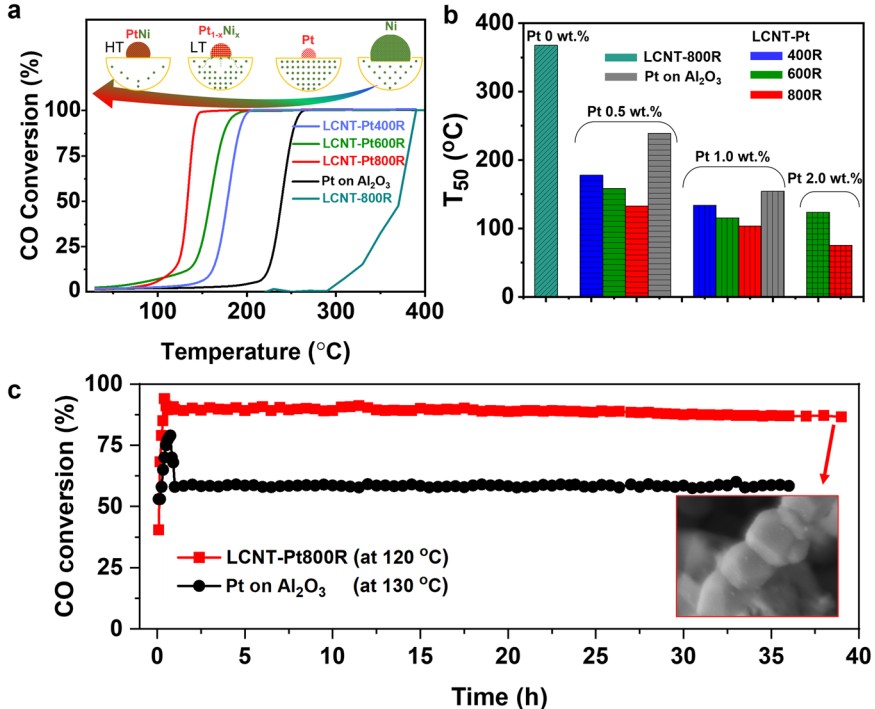

**Fig. 5 | Catalytic functionality on CO oxidation reactions at different conditions for the perovskite fibres catalysts with in-situ Ni decorated Pt nanoparticles.** Catalytic CO oxidation performances on (**a**) light-off curves of LCNT, LCNT-Pt fibres (Pt 0.5 wt.%) reduced at 400 °C, 600 °C, 800 °C comparing with LCNT and Pt on γ-Al$_2$O$_3$ catalysts as a reference, and (**b**) $T_{50}$ values (the temperature at 50% CO conversion) summarised from Supplementary Fig. 18b, c for the prepared samples with different Pt loading (0.5, 1.0, 2.0 wt.%) of reduced LCNT-Pt fibres and Pt on γ-Al$_2$O$_3$ catalysts, as well as (**c**) long-term tests through the catalytic time on stream method with constant temperatures at $T_{90}$ of 120 °C for LCNT-Pt800R (Pt 0.5 wt.%) and 130 °C for Pt on γ-Al$_2$O$_3$ (1.0 wt.%), respectively. All catalytic experiments were carried out by a feed mixture gas of 20,000 ppm CO, 10.0 vol.% O$_2$ from the air (21 % O$_2$ and 79 % N$_2$) at N$_2$ balance with a total gas flow rate of 200 ml/min (GHSV = 60,000/h).

performance shows a considerably high TOF$_{150}$ value of 0.489 s$^{-1}$ at low surface area (2-3 m$^2$/g, see Supplementary Fig. 19), which is much greater than that of the commercial type Pt on γ-Al$_2$O$_3$ catalysts of 0.031 s$^{-1}$ (0.5 wt.%, 150 m$^2$/g, the morphology was shown in Supplementary Fig. 20) even at higher Pt loading of 0.295 s$^{-1}$ (1.0 wt.%). As shown in Fig. 5b, the light-off points of $T_{50}$ for our catalysts were further improved at higher Pt loading with an outstanding catalytic activity, which should come from the active metal alloy structure with a stimulated perovskite oxygen vacancy through the in-situ Ni decoration to Pt for highly distributed small nanoparticles.

The stability of the catalytic activity of the LCNT-Pt sample was investigated by setting the temperature to achieve a 90% conversion point ($T_{90}$) at the onset of ageing ($T_{90}$). The conversion ratio was fairly stable for around 40 h with a slight drop at the initial one hour (Fig. 5c). Differently, the commercial Pt on Al$_2$O$_3$ catalysts showed fast degradation after stabilising at a similar reaction temperature, and thereafter with a steady conversion ratio of 58% due to the lower reactivity. In addition, the durability tests for the LCNT-Pt samples were carried out by maintaining at $T_{99}$ (Supplementary Fig. 18d). The results indicate no physical degradation for all these samples for 15 h testing at high conversion rates, which revealed high stability. It was found that our catalysts demonstrate a good tolerance against the coalesce after the light-off and long-term tests retaining the nanofibrous structure with size and dispersion of the nanoparticles on perovskite (Supplementary Fig. 21), which indicates the catalytic performance stability thus relates to the structural stability of the catalyst even under oxidation conditions for such a long period (embed image in Fig. 5c). The interaction between metal and support affords stabilisation of small metal nanoparticles[48,49], moreover, increasing surface area means the there is more surface for an exsolved material to rest upon and so will tend to reduce particle size. At high temperatures high surface area

also contributes to the sintering suppression by increasing the dispersion[50]. Note that, typically a perovskite support is constrained in its metal dispersion as high crystallisation temperature results in low surface area[51]. Thus three metal dispersion can be enhanced by nano-architecture engineering, such as the fibre structure used in this work.

## Discussion

In summary, we report an in-situ dynamic formation of highly distributed small Ni-Pt bimetallic nanoparticles via reactive metal-support interaction, which can also be applied to other systems. The composition and structure of the bimetallic nanoparticles were efficiently tuneable in the perovskite by controlling the exsolution of cations from the A-site deficient perovskite system and by using a surface additive metal as a promoter. A robust catalyst of Pt-Ni nanoparticles on fibres yielding a catalytically efficient configuration in various CO oxidation tests through an active and stable nanoparticulate structure with good metal support interactions, demonstrating a new paradigm for nanoparticle design in wide variety of important applications including catalytic reaction systems.

## Methods

### Materials synthesis and Pt deposition

Lanthanum chloride heptahydrate (LaCl$_3$·6H$_2$O, 99 %, Alfa Aesar), Calcium chloride (CaCl$_2$, Fisher Scientific), Nickel chloride hexahydrate (NiCl$_2$·6H$_2$O, 99.9 %, Alfa Aesar), Titanium (IV) isopropoxide (C$_{12}$H$_{20}$O$_4$Ti, 97 %, Aldrich), H$_2$[Pt(NO$_3$)$_6$] (16.7 %, nitrate stabilised, Johnson Matthey) and Polyvinylpyrrolidone (M.W. = 1,300,000, Alfa Aesar) were used as raw materials. All these reagents were used as received without further treatments.

La$_{0.52}$Ca$_{0.28}$Ni$_{0.06}$Ti$_{0.94}$O$_3$ (LCNT) fibres were prepared via the electrospinning method, the detail can also be seen on our previous

work[52]. The diagram presented in Supplementary Fig. 1 illustrates the process for preparation of the perovskite fibre support. In a typical procedure for electrospinning, stoichiometric amounts of $LaCl_3$ $6H_2O$, $CaCl_2$, $NiCl_2$ $6H_2O$, Titanium (IV) isopropoxide and PVP were dissolved in ethanol and acetic acid mixed solvent (volume ratio 2:1). The acetic acid stabilises the hydrolysis reaction of the sol-gel precursor. The solution was stirred for 2 h at ambient temperature to form the homogeneous precursor required for electrospinning. The concentration of the LCNT perovskite was 0.2 M in the precursor solution. The PVP in the metal solution was kept at 2.5 wt% to reach an optimized viscosity. The solution was then loaded into a plastic syringe (5 mL) with a stainless-steel spinneret (D ¼ 0.5 mm). The supply rate of the precursor in the electrospinning set-up was set as 0.3 $mLh^{-1}$ with a 12 kV voltage was applied on the spinneret, with the fibre collected on grounding aluminium foil collector at a distance of 10 cm. The electrospun nanofibres were collected and dried at 80 °C for 5 h followed by calcination at 1100 °C for 2 h to achieve a pure perovskite phase. The as-prepared LCNT was further treated in 5 % $H_2$/Ar for 4 h at 400 °C, 600 °C and 800 °C to obtain the reduced perovskite samples. Perovskite fibre without Ni doping, i.e., $La_{0.52}Ca_{0.28}TiO_3$ was also prepared via the same electrospinning procedure.

The Pt species was introduced onto the fibre perovskite support by deposition-precipitation using $H_2[Pt(NO_3)_6]$ (0.0778 g) dissolved in 50 ml water. The Pt loading amount was 0.5 wt% on the perovskite fibre (100 mg). The Pt solution (~ 1.708 $mmol·L^{-1}$, 2 ml) was applied to wet the perovskite fibre followed by drying at 80 °C in air for 10 h to form the active catalyst material. This was then calcined at 400 °C for 2 h to remove the nitrate. As a comparison, the same procedure was carried out with different Pt loading of 1.0 wt% and 2.0 wt%, on high surface area γ-$Al_2O_3$ (sigma, surface area 150 $m^2·g^{-1}$). The perovskite pellet with the same composition also underwent this deposition procedure. Noted that the only exposed surface dense pellet bring difficulty of determine the loading amount of Pt for this situation.

## Hydrogen reduction and process characterisation

The as-prepared LCNT was further treated in 5 % $H_2$/Ar for 4 h at 400 °C, 600 °C and 800 °C in a tube furnace to obtain the reduced perovskite samples. The preparation of Pt-LCNT reduced samples was carried out by treating the samples at three different temperatures, denoted as LCNT_0.5Pt800R, LCNT_Pt600R and LCNT_Pt400R, respectively.

TGA measurements were performed on a NETZSCH STA 449 C instrument using Proteus thermal analysis software. The initial weight of the sample is about 40 mg. The mass was recorded when heated to 800 °C under flowing 5% $H_2$/Ar (30 $ml·min^{-1}$), with a heating rate of 5 °$C·min^{-1}$. Thereafter, an isothermal procedure was carried out for 10 h at 800 °C. The blank with an empty crucible was recorded under the same conditions and subtracted from the sample data. The raw data were numerically differentiated to obtain differential thermogravimetry curves.

TPR was carried out using a Micromeritics AutoChem II 2920 instrument with about 100 mg samples. The sample was loaded in the quartz glass reactor and purged with He at 50 $ml·min^{-1}$ flow rate, at 400 °C for 1 h and cooling down to 50 °C to remove any trace of adsorbates. Then, TPR was conducted in 10 % $H_2$/Ar at a flow rate of 50 $ml·min^{-1}$. The temperature raised from room temperature to 1000 °C with a heating rate as 10 °$C·min^{-1}$.

## Basic characterisations

XRD measurements were performed at room temperature angle from 10-90 ° 2θ using a PAN analytical Empyrean diffractometer with Cu Kα1 radiation (1.54056 Å) and Bragg-Brentano geometry operated in reflection mode.

The XPS spectra were recorded on a Thermo Scientific K-Alpha⁺ X-ray photoelectron spectrometer operating at $2 × 10^{-9}$ mbar base pressure. This system incorporates a monochromated micro-focused Al Ka X-ray source ($hυ = 1486.6$ eV) and a 180 ° double-focusing hemispherical analyser with a 2D detector. The X-ray source was operated at 6 mA emission current and 12 kV anode bias providing an X-ray spot size of up to 400 $mm^2$. Survey spectra were recorded at 200 eV pass energy 20 eV pass energy for core level spectra, and 15 eV pass energy for valence band spectra. A flood gun was used to minimise the sample charging that occurs when exposing an insulated sample to an X-ray beam. The binding energy of each sample was corrected by the adventitious carbon C-C located at 284.8 eV. The quantitative XPS analysis was performed using Avantage software.

To confirm the loading percentage of platinum on support, we used inductively coupled plasma optical emission spectrometry (Thermo Fisher Scientific ICP-OES iCAP 6000 Series) to analyse the amount of platinum. Samples (20 mg each) were dissolved into aqua regia at 80 °C and left overnight to dissolve the sample. Though it is hard to digest the perovskite completely, the treatment should be enough to dissolve the platinum coated on the surface. The measurement used a diluted solution. The 0.5 wt% Pt loaded sample should display a concentration of 10 ppm ($μg·mL^{-1}$). The average data indicate that the loading amount of platinum in the samples is within the measuring error limits of the designed stoichiometric values, as shown in Table S1.

Specific surface area ($S_{BET}$), average pore volume and pore size of the prepared samples were determined by the distribution graph of $N_2$ adsorption-desorption at 77 K using Micromeritics TRISTAR II 2020. Samples are outgassed at 200 °C under vacuum for 12 h using a Quantachrome Flovac degasser (Micromeritics VacPrep 061). The $N_2$ isothermal data set was collected at -195 °C. $S_{BET}$ is calculated by the Brunauer-Emmett-Teller multiple-point method at a partial pressure range from 0.05 to 0.3. Total pore volume is determined at $p/p_0 = 0.99$.

The X-ray absorption near edge structure (XANES) and extended X-ray absorption fine structure (EXAFS) of the Ni K (8.3330 KeV) edge and Pt $L_3$ (11.5634 KeV) edge for selected samples were collected at ambient temperature, on the B18 station at Diamond Light Source national synchrotron facility, UK. The absorption near the Ni K edges was determined from the total transmission yield and Pt $L_3$ edges were determined from the total fluorescence (with a nine-element germanium detector) yield obtained with linearly polarised photons. Measurements were carried out using a Si(111) monochromator at Pt $L_3$-edge with a Pt and Ni monometallic foil (10 μm) used as an energy calibrant before measurement. Samples were pressed into 13 mm pellets, and high Pt-loaded samples were diluted using an appropriate amount of cellulose binder. Pt foil and $PtO_2$ were used as references for $Pt^0$ and $Pt^{4+}$, respectively. The data were analysed using Athena from the Deterter suite, which implements the FEFF6 and IFEFFIT codes and Artemis[53,54]. The spectra were background-corrected and normalised by subtracting a linear fit and by dividing the post-edge region by a third-degree polynomial fit.

## Microstructure characterisation

A FEI Scios electron microscope equipped with secondary and back-scattered electron detector was used to acquire high-resolution images for investigating the morphology of nanofibres and the exsolution of nanoparticles. The number and diameter/size of the fibres/NPs were calculated from the magnified images using Image-Pro Plus software. The diameter of fibres and size of particles were calculated by a calibration of the SEM image scale between pixel and nm, where the particles are assumed to be hemispheric.

To analyse the size and characteristics of the NPs on the sample, high-angle annular dark field (HAADF) STEM images were taken with an FEI Titan Themis instrument, using a 25 keV beam with 0.2 pA beam current. The specimens for STEM were first ground in acetone before loading on the copper mesh. To investigate the topological

and elemental information of nanoparticles, a thin lamella was prepared via a focused ion beam (FIB). This technique allows us to analyse the interface via checking the cross-section of fibres. Supplementary Fig. 7 presents a general procedure to prepare TEM specimens by FIB with fibre samples. The fibres were firstly dispersed with acetone and then sprinkled on a conducted silicon wafer piece. The fibre was selected under high-resolution SEM, then coated with carbon as a protective layer, as shown in the black area of Supplementary Fig. 8–11. The specimen was polished by an ion beam to achieve a certain thickness (below 100 nm). Finally, the coated fibre is cut with FIB and loaded on the TEM grid with a tungsten needle. High-resolution STEM images of individual nanoparticles were recorded to study the shape of particles and interface. The energy dispersive X-ray (EDX) data were also extracted by line and mapping analysis to distinguish the composition of NPs on the sample surface compared to the desired perovskite.

## Catalytic performance test

The CO oxidation performances of samples were evaluated in a laboratory-scale fixed–bed quartz reactor (internal d8 mm) under atmospheric pressure with a similar procedure as described in our previous work[19]. Typically, about 100 mg of samples were equipped in the reactor by quartz wool at each side. The temperature was recorded by the K-type thermocouple at the centre of the central reaction zone and gas flows were controlled by the electronic mass-flow controllers (MFC, ALICAT scientific). The samples were pre-treated in situ with 5% $H_2$/Ar at 400 °C for 1 h, and then purged with $N_2$ for 30 min to remove the residual $H_2$. After cooling to 20 °C, a feed mixture gas of 20,000 ppm CO, 10.0 vol.% $O_2$ from the air (21% $O_2$ and 79% $N_2$) at $N_2$ balance is introduced with a total gas flow rate of 200 ml·min$^{-1}$ (GHSV = 60,000 h$^{-1}$). The light-off experiments were measured at the range of 20–400 °C with a ramp rate of 5 °C·min$^{-1}$. The composition of the effluent gas products was measured by online TCD gas chromatography (GC-2014, SHIMADZU) equipped with a molecular sieve 5 A (60–80 mesh) column. The CO conversion was calculated from the change in the CO concentrations as follows:

$$CO\ converison(\%) = \frac{[CO]_{In\ Vol\%} - [CO]_{out\ Vol\%}}{[CO]_{In\ Vol\%}} \times 100 \quad (1)$$

The specific reactivity was compared through the estimated TOF[55] values which were calculated by utilising a geometrically based model with the assumed surface area of Pt(-Ni) hemisphere nanoparticles.

$$TOF_{150} = \frac{Conv_a \times F_{CO}}{S_a} \quad (2)$$

$TOF_{150}$ is the turnover of frequency at 150 °C, $Conv_a$ is the conversion in % at 150 °C, $S_a$ is the number of active sites (0.1 g catalyst x µmol dispersed Pt-(Ni) phase per gram catalyst) and $F_{co}$ is the flux rate of CO (4 ml·min$^{-1}$, so 2.725 µmoles·s$^{-1}$).

The long-term tests were carried out at both 90 % and 100 % conversions. For the 90 % conversion stability test, the temperature was set at 120 °C for the samples of Pt-LCNT-800R and Pt on $Al_2O_3$, respectively, sampling initially every 10 min for 1 h and then every 30 min for more than 40 h. Moreover, the stability tests under conversion conditions were carried out from the initial CO oxidation conditions with constant temperatures at each 100 % conversion points of 160 °C and 130 °C for the samples of Pt-LCNT-600R and -800R, respectively, sampling every 20 min for more than 15 h.

## Data availability

The research data underpinning this publication can be accessed at https://doi.org/10.17630/8bc13ae7-f04d-4a26-8648-54929bc5fcc4.

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

## Acknowledgements

We thank the Diamond Light Source and Prof. Alan Chadwick for the award of beam time as part of the Energy Materials Block Allocation Group SP14239;

## Author contributions

M.X. and Y.K.J. designed and carried out the experiments, and wrote the manuscript. A.N. collected TEM and EDX data. H.K. helped with the catalytic performance test. G.K. and D.J.P. collected XPS data. Y.S. discussed the idea of this work. J.T.S.I. supervised the whole study and revised the manuscript. All authors discussed the results and commented on the manuscript.

## Funding

EPSRC for a Critical Mass project EP/R023522/1 and Electron Microscopy provision EP/R023751/1, EP/L017008/1.

## Competing interests

The authors declare no competing interests.
