## [Peer Review File · Nature Communications]

Synergistic Growth of Nickel and Platinum Nanoparticles via Exsolution and Surface ReactionReviewers' comments:

Reviewer #1 (Remarks to the Author):

The authors addressed most previous comments on the paper reasonably.

However, it is important to point out that regardless of whether others have published this sort of stability analysis, it does not actually assess stability. Consider if 1 g of catalyst is used in a reaction for which 0.1 g reaches 100% conversion. Then catalyst deactivation could happen for 90% of the catalyst before it is detectable. Thus, rather than following an approach used in the field that is not effective, the authors should consider doing this experiment in a reasonable way - starting well below 100% conversion.

Reviewer #2 (Remarks to the Author):

In the review of the first version I have provided 19 concrete comments on what should be changed/improved. Of those 13 comments (7-19) were technical/scientific in nature. In most cases the responses are unsatisfactory and sometimes not scientific. As an example I note the rebuttal to my comment 17 (that is the same as comment 4 of reviewer 1). The authors cite a number of papers that have made the same mistake as themselves: measuring at 100% conversion is not correct to assess stability. Please note that the references put forward are all to non-catalysis journals.

Dear editor and reviewers,

The authors appreciate the editor and reviewers for providing valuable comments and suggestions that improve the quality of our manuscript. Based on those comments, we have carefully revised the manuscript. The modified text in the revised manuscript was highlighted. Please see below for our detailed point-to-point responses.

Reviewer #1

General Comment

The authors addressed most previous comments on the paper reasonably. However, it is important to point out that regardless of whether others have published this sort of stability analysis, it does not actually assess stability. Consider if 1 g of catalyst is used in a reaction for which 0.1 g reaches 100% conversion. Then catalyst deactivation could happen for 90% of the catalyst before it is detectable. Thus, rather than following an approach used in the field that is not effective, the authors should consider doing this experiment in a reasonable way - starting well below 100% conversion.

Response to General Comment

Whilst we were focusing on physical stability, the referee's suggestion about reducing conversion to assess catalytic activity stability adds an important new dimension to our study. Therefore as suggested by the reviewer, catalytic stability analysis below 100% conversion has been carried out. We revisited this work and carefully addressed stability from the catalytic viewpoint rather than the materials viewpoint. This stability test was carried out around 90% conversion for about 40 h, as shown in Fig.5c and below.

Figure 5. Catalytic functionality on CO oxidation reactions at different conditions for the perovskite fibre catalysts with *in-situ* Ni decorated Pt nanoparticles. Catalytic CO oxidation

performances on (a) light-off curves of LCNT, LCNT-Pt fibres (Pt 0.5 wt.%) reduced at 400 °C, 600 °C, 800 °C comparing with LCNT and Pt on γ -Al₂O₃ catalysts as a reference, and (b) T_{50} values (temperature at 50% CO conversion) summarized from Fig.S16b-c for the prepared samples with different Pt loading (0.5, 1.0, 2.0 wt.%) of reduced LCNT-Pt fibres and Pt on γ -Al₂O₃ catalysts, as well as (c) long-term tests through the catalytic time on stream method with constant temperatures at T_{90} of 120 °C for LCNT-Pt800R (Pt 0.5 wt.%) and 130 °C for Pt on γ -Al₂O₃ (1.0 wt.%), respectively. All catalytic experiments were carried out on a feed mixture gas of 20,000 ppm CO, 10.0 vol.% O₂ from air (21 % O₂ and 79 % N₂) at N₂ balance with a total gas flow rate of 200 ml/min (GHSV=60,000/h).

Description below was added in the text:

The stability of the catalytic activity of the LCNT-Pt sample was investigated by setting temperature to achieve 90% conversion point (T_{90}) at the onset of ageing (T_{90}). The conversion ratio was fairly stable for around 40 h with a slight drop at the initial one hour (Fig.5c). Differently, the commercial Pt on Al₂O₃ catalysts show fast degradation after stabilizing at similar reaction temperature, and thereafter with a steady conversion ratio of 58% due to the lower reactivity.

The morphology of the used catalyst exhibits good structural stability even after long operation test (Fig.5c and Fig.S18). The 100% conversion stability data were moved to supporting information in Fig.S16.

Reviewer #2

General Comment

In the review of the first version I have provided 19 concrete comments on what should be changed/improved. Of those 13 comments (7-19) were technical/scientific in nature. In most cases the responses are unsatisfactory and sometimes not scientific. As an example I note the rebuttal to my comment 17 (that is the same as comment 4 of reviewer 1). The authors cite a number of papers that have made the same mistake as themselves: measuring at 100% conversion is not correct to assess stability. Please note that the references put forward are all to non-catalysis journals.

Response to General Comment

Apart from 17, the referee's comments are not really specific here about what is unsatisfactory and so it is hard to respond to these. We have attempted to clarify the response to 7-19 mentioned, hopefully we have understood the nature of concerns, please see below.

The referee comments seem a little catalysis focused. Certainly catalysis is very important and we are keen to apply our approaches in this domain; however, the key advances are in the nanoparticle formation and support interaction. This can relate to catalysis but we see the work as much more generic and enabling. In our initial submission we claimed that nanomaterials were stable on long term testing, we did not claim stable catalytic activity. It therefore seems disputable to say that this is a mistake as stated above by the referee. We take the point that catalytic activity is best assessed at below 100% conversion and indeed performing such an experiment considerably adds to the study, showing that not only are the nanoparticles physically stable, their activity is also stable. As detailed above we have been able to perform additional tests and acknowledge that these have been very worthwhile.

For the comment 17, the stability test was carried out below 100% conversion (around 90%) for about 40 h to examine the catalytic stability, now shown in Fig.5c. The conversion ratio was stable for 40 h after a slight drop in the initial hour. At a similar temperature (but 80% conversion initially), the commercial Pt on Al₂O₃ catalysts show fast degradation after stabilizing at similar reaction temperature, and thereafter with a steady conversion ratio of 58% due to the lower reactivity.

Revised response to previous comments

Reviewer #2
Comment 7. The support material used for this study (LCNT) has a very low specific surface area, 2-3 m ² /g. This makes the resulting catalysts not generally useful for catalysis. The authors should be more explicit about this. Also that high surface area supports of say 200 m ² /g can stabilize small noble metal particles in a more straightforward manner.
Response to Comment 7 The objective of the nanofiber support was to increase surface area of the support which it did by a factor of 2 or 3. Clearly this was not as much as we would have liked; however performance and durability exceeds that of nanoparticles deposited more conventionally on high surface area supports. This is strong justification for the current study and leaves the huge potential of stable exsolution from supports with surface areas of 200m ² /g. A major consideration from these studies is that conventional deposition yields dispersions that are much less robust than those that are exsolved out of the support. The term “stabilise” in “high surface area supports of say 200 m ² /g can stabilize small noble metal particles in a more straightforward manner” seems incorrect, the word should be “support”
Comment 8. Pt nanoparticles were dispersed on LCNT by ‘careful deposition’ (page 4). Much to my surprise the manuscript (main text plus SI) does not contain sufficient details about the ‘deposition precipitation’ (SI) – what was the precipitating agent, what time, concentrations etc were applied. It should be made possible for readers to reproduce the synthesis.
Response to Comment 8 Thank you for alerting this. We highlight the details about our deposition process. The details included in the supporting information part as ‘Pt deposition on perovskite fibers’, a H ₂ [Pt(NO ₃) ₆] (16.7 %, nitrate stabilized, Johnson Matthey) was used as Pt resource and diluted in deionized water, with drying at 80 °C for 10 h to allow the evaporation of water. To make it easier for readers to reproduce, we add the weight of perovskite fiber (100 mg) and concentration of precursor (~1.708 mmol/L) in the text.
Comment 9. The authors speak about ‘precursor diffusion’ and ‘uneven wetting’ (page 4) as important aspects of preparation. This is difficult to understand without details of the deposition precipitation but not generally true.
Response to Comment 9 It is surprising to hear that “uneven wetting is not generally true”, this may be so if one restricts to a very narrow range of substrates and solutions as might be expected in typical heterogeneous catalysis systems, but is certainly not the case when one looks at wide ranges of substrates as one does in energy materials. The detailed deposition method was added in the ‘materials synthesis and Pt deposition’ part. The ICP results (table S1) indicating the even deposition of Pt on perovskite fibers was added for proof of deposition amount. The diffusion of precursor and wetting condition were compared with various materials, including exsolved materials and pellets. The morphology of those samples shows either slightly uneven distribution or etched surface (see Fig.S5 and Fig.S6).
Comment 10. Page 4: “The SEM micrograph in Fig. 1d shows well-dispersed nanoparticles”, however, Figure S6 shows particles as large as 14 nm, so far from well-dispersed.
Response to Comment 10 These images are from very different samples so the larger particle size is not a matter of conflict, The large particles are for a dense pellet reduced at 800°C and the small particles are on a reduced

fibre sample. A key point is that the fibre provides a better surface than the dense pellet for nanoparticle creation,

The SEM micrograph in Fig.1b shows well-dispersed and much smaller nanoparticles decorated on the surface of perovskite fibres with Pt than without Pt (also see Fig.S4 for comparison with other samples). With the same deposition procedure applied, small particles formed on the surface even for a pellet sample (Fig.S5). The slightly larger particle size may relate to uneven wetting on different orientations and differences in surface composition of the perovskite as A-site enrichment is more significant in lower surface area bodies. This highlights the advantage to achieving more small nanoparticles on the fibre framework,

Comment 11. Page 8, last paragraph: strain distribution in the nanoparticles. Please discuss the state of the nanoparticles: fully reduced or reduced and passivated. Details about the sample preparation for TEM study needed.

Response to Comment 11

Thank you for the comment.

The XAS and XPS results that indicates the Pt species are in metallic state when reduced at temperature above 400 °C. While the Ni species present mainly at high oxidation states which was only reduced with increasing the temperature. Moreover, the EDX mapping results show nanoparticles in LCNT_Pt600R and LCNT_Pt800R are all in metallic states, i.e. oxide free, which see the following figures.

Figure S12. Representative HAADF-STEM images and EDX analysis of LCNT-Pt nanofiber catalyst reduced at 800 °C for 4 h in 5% H₂/Ar.

Figure S9. Representative HAADF-STEM images and EDX analysis of LCNT-Pt nanofiber catalyst reduced at 600 °C for 4 h in 5% H₂/Ar.

The detail of sample preparation from TEM were added in the SI. In detail, the samples were reduced at different temperature (400 to 800 °C) then cooling down to room temperature in 5% H₂/N₂. Thus, the state of nanoparticles should be referred to as reduced.

The cross-section of fibre specimens was prepared by focused ion beam (FIB) milling to understand the underlying mechanism for the formation of nanoparticles. Fig.S8 presents a general procedure to prepare TEM specimens by FIB with fibre samples. The fibres were firstly dispersed with acetone and then sprinkled on a conducted silicon wafer piece. The objected fibre was picked under high resolution SEM, then coated with carbon as a protective layer, as shown in the black area of Fig.8-9(d). The specimen was polished by an ion beam to achieve a certain thickness (below 100 nm). Finally, the coated fibre is cut with FIB and loaded on the TEM grid with a tungsten needle.

Figure S8. Flow chart of TEM fibre sample prepared by FIB on Scios Dual-beam platform.

Comment 12. Page 11, about the mechanism how Pt catalyses extraction of nickel. Rather than Pt pulling our cations I would suggest that Pt facilitates H₂ dissociation and thus reduction of the Ni cations.

Response to Comment 12

Pt promotes surface exchange of oxygen and so enables reduction of the lattice, this in turn facilitates Ni migration as this needs to be balanced by oxide ion and electron migration. This is normally how the influence of Pt is noted in solid oxide type systems. This could easily be considered as analogous to the above mentioned H₂ dissociation as the processes are allied.

Please see the text added to the manuscript:

Thermogravimetric analysis (TGA) shows more extensive and facile weight loss under hydrogen atmosphere for the sample with Pt loading than the unloaded perovskite (Fig.1c). The onset point shifts to lower temperature by about 70 °C, with about 40 % more weight loss for the Pt loaded samples until the temperature reached 800 °C. This shows more oxygen vacancy formation (0.111 vs 0.085 on reduction at 800 °C for 10 h) under the same treatment conditions most likely due to enhanced surface exchange due to the presence of Pt.

Comment 13. The catalysis experiments are described in SI but lack essential details. What grain size (sieve fraction) of the catalysts have been used? Powder cannot be used to obtain reliable results.

Response to Comment 13

The alumina based sample were sieved to commercial size whereas the perovskite fiber was used as catalyst without sieving, see the morphology in Fig.1 and Fig.S1, S5. The used catalyst was piece of nonwoven fibers, with size around hundreds of micro-meters, as shown in Figure 4.

Figure 4. Low magnification SEM of (a) fiber catalyst and (b) Pt/Al₂O₃ catalyst used in this work.

Noted that, the fiber morphology is fairly robust as shown in an electrochemical study when used as electrode in solid oxide fuel cells (Figure 2).

Figure 5. LCNT fiber as electrode for fuel cell after grounding for 1 hour and calcined on electrolyte at 1100 °C. (Data was published in *J. Mater. Chem. A* 2023, 11, 13007-13015.)

Comment 14. Catalysis measurements have been done with a ramp of 5 C/min. Going up in temperature? If so then also data for cooling down should be provided to reveal possible hysteresis.

Response to Comment 14

The heating ramp rate used in CO oxidation test is $5\text{ }^{\circ}\text{C}\cdot\text{min}^{-1}$. Interestingly, the LCNT-Pt800R catalyst demonstrates a self-sustained behaviour during the cooling process with CO conversion close to 99%. A hysteresis phenomenon was widely reported for CO oxidation contributing to the hotspot in reactor (*Catalysts* 2018, 8 (12)). However, the dynamic structural changes of catalyst and should take into account as it is not fully credible as a long self-sustention Pt-LCNT-600R fibres, as shown in Fig.S17.

Figure 6. CO oxidation test with heating and cooling with LCNT-Pt800R as catalyst.

The Pt-LCNT-600R fibres catalyst possesses high reactivity with over 80 % CO conversion even at room temperature. The low temperature catalytic performance was also maintained over 15 h. A proper investigation to understand the self-sustained phenomena will be carried out for future exploration.

Fig.S17(c) The catalytic time term test at room temperature for the prepared sample of LCNT-Pt nanofiber reduced at 600 °C.

Comment 15. TOF data have been provided based on the amount of metal used. However, by definition, TOF data should be based on the number of surface sites, not on total metal.

Response to Comment 15

Thank you for your comment, apologies for lack of clarity. Yes of course TOF is based upon surface area of catalyst. We meant that we assumed the active surface was the available surface of the metal nanoparticles excluding the substrate and interfacial layers. This may not be fully correct in the case of these perovskites which may have some activity, but this is best approach

To evaluate the TOF in this work, we use a geometrically based model and considered of the interface of nanoparticles. The TOF was estimated with the surface area of the surface particles by assuming the hemisphere structure (Nat. Chem. 2021, 13, 677–682. and J. Mol. Catal., A: Chem. 374–375 (2013) 53– 58.) The details of this analysis and comparisons are included in the supplementary information.

Comment 16. Catalysts after CO oxidation catalyst should be studied by TEM to reveal possible particle growth.

Response to Comment 16

Thank you for the suggestion.

The morphology of the sample after stability test was studied, the picture shows slight growth of particles under oxidation atmosphere at high temperature up to 400 °C. However, a well dispersion and small size were maintained even after 40 h stability test at 90 % conversion, as shown in Fig.S18. This suggests the catalytic performance stability thus relates to the structural stability of the catalyst even under oxidation conditions for such a long period.

Figure S18. SEM images of LCNT-Pt800R (Pt 0.5 wt%) nanofiber catalysts after (a) CO light-off test to 400 °C and (b) long-term CO oxidation test for 40 hours. All experiments were carried out by a feed mixture gas of 20,000 ppm CO, 10.0 vol.% O₂ from air (21% O₂ and 79% N₂) at N₂ balance with a total gas flow rate of 200 ml/min (GHSV=60,000/h).

Comment 17. The durability experiments (Fig. 5b) do not suffice for assessment of stability since conversion was 100%. Please study and report stability at for example 50 conversion.

See above and in response to reviewer 1

Comment 18. Provide information about the commercial Pt/alumina catalysts, in particular Pt nanoparticle size.

Response to Comment 18

Thank you for the suggestion.

The Pt/ γ -Al₂O₃ (sigma, surface area 150 m²/g) catalyst was prepared by a deposition-precipitation method using H₂[Pt(NO₃)₆] dilute in water follow the same procedure as LCNT-Pt samples. In detail, the Pt loading amount was set as 0.5 wt% of the weighted perovskite fiber (100 mg). The Pt resource (~1.708 mmol/L) was used to immerse perovskite fiber followed with drying at 80 °C in air for 10 h to form the as-prepared active catalyst material, then calcined at 400 °C for 2 h to remove the nitrate. The particles morphology can be seen from our previous work where the sample was prepared under the same procedure.

Figure 7. NPs analysis after long-term aging tests. (Data was published in Nat. Chem. 2021, 13, 677–682.)

TEM of the commercial Pt on γ -Al₂O₃ catalysts used in this work was measured to identify the morphology of the Pt nanoparticles, as shown in Fig.S18. The Pt loaded on high surface γ -Al₂O₃ exhibit small particle size from 0.5 nm to 2.5 nm, with an average size about 1.8 nm. Based on the TEM of Pt/Al₂O₃, the Pt particles have average size about 1.8 nm, which is close to the LCNT-Pt samples, as shown in Fig.2.

Figure S18. TEM images of commercial Pt on γ -Al₂O₃ catalysts (0.5 wt.%, 150 m²/g) at different sampling region. The embed figure shows the size distribution of Pt nanoparticles.

Comment 19. The low TOF of the commercial Pt/Al₂O₃ catalysts could be caused by a very low Pt nanoparticle size (1 nm) that leads to too strong oxygen bonding thus lowering CO oxidation activity. I challenge the authors to pre-treat that catalyst at say 600C such that Pt nanoparticles will grow to say 3 nm and then these might display much higher activities.

Response to Comment 19

We agree that the particles in our Pt/Al₂O₃ samples might be smaller than optimal, but we also tested commercial samples with 2nm particles and findings were similar. We compared the commercial Pt/Al₂O₃ with 1 wt% Pt loading as shown in Fig.S16(a). The TOF increased with more Pt loading amount. The CO conversion at T₅₀ also improved with more Pt loaded for Pt/Al₂O₃ sample. However, it is still inferior to our samples, as shown in Fig.5(b) and Fig.S16. The T₅₀ of the catalytic test was compared with different Pt loading indicating the reactivity of material increased when possess more active sites.

Fig.S16(a). calculated TOF_{metal} (s^{-1}) values of samples involved in this work.

Fig.5(b). T_{50} values (temperature at 50 % CO conversion) summarized from Fig.S16b-c for the prepared samples with different Pt loading (0.5, 1.0, 2.0 wt%) of reduced LCNT-Pt fibres and Pt on γ -Al₂O₃ catalysts

The performance can be compared with the Pt/ γ -Al₂O₃ samples (prepared with same procedure) which were reduced in 5% H₂/Ar flow for 12 h at 500 °C.

Figure 8. Light-off curves for the initial CO oxidation for a feed mixture gas of 20,000 ppm CO, 10.0 vol% O₂ from air (21% O₂ and 79% N₂) at N₂ balance with a total gas flow rate of 200 ml min⁻¹ (GHSV = 60,000 h⁻¹) on the prepared samples. The Pt/Al₂O₃ were highlight with purple frame and arrow (Data was published in Nat. Chem. 2021, 13, 677–682.)

Note that we have described how the catalytic performance was evaluated in SI, as 'with the samples were pre-treated in-situ with 5% H₂/Ar at 400 °C for 1 h, and then purged with N₂ for 30 min to remove the residual H₂. After cooling to 20 °C, a feed mixture gas of 20,000 ppm CO, 10.0 vol.% O₂ from air (21% O₂ and 79% N₂) at N₂ balance is introduced with a total gas flow rate of 200 ml·min⁻¹ (GHSV=60,000/h)'.

Therefore, we could say the performance of reduced Pt-perovskite still surpass the Pt/Al₂O₃ under the treated conditions.

REVIEWER COMMENTS

Reviewer #1 (Remarks to the Author):

The authors reasonably addressed the comments. The paper can be published.

Reviewer #2 (Remarks to the Author):

The authors have now addressed more issues. Still I feel that my comment 7 (that a high-surface-area support can readily be used to stabilize 3 nm metal particles) should be added. Without this the paper might be confusing to the average reader of NCOMM.

Dear editor and reviewers,

The authors thank the editor and reviewers for providing valuable comments and suggestions that improve the quality of this manuscript over the entire process. Based on the comments, we have appropriately revised the manuscript. Please see below for our detailed responses to each point. The modified text was highlighted in the revised manuscript.

Reviewer #1
General Comment The authors reasonably addressed the comments. The paper can be published.
Response to Comment Thank you for your positive comment.

Reviewer #2
General Comment The authors have now addressed more issues. Still I feel that my comment 7 (that a high-surface-area support can readily be used to stabilize 3 nm metal particles) should be added. Without this the paper might be confusing to the average reader of NCOMM.
Response to Comment 1 Thank you for the suggestion. We have added discussion about the metal particle size stabilization function of high-surface area supports. The possible limitations of the catalyst with small surface area were also discussed in the text. “The interaction between metal and support affords stabilisation of small metal nanoparticles (48, 49), moreover, increasing surface area means the there is more surface for an exolved material to rest upon and so will tend to reduce particle size. At high temperature high surface area also contributes to the sintering suppression by increasing the dispersion(50). Note that, typically a perovskite support is constrained in its metal dispersion as high crystallization temperature results in low surface area(51). Thus thee metal dispersion can be enhanced by nano-architecture engineering, such as the fiber structure used in this work.”